# Robust Watermarking for Diffusion Models: A Unified Multi-Dimensional Recipe

## Abstract

Diffusion models are known for the supreme capability to generate realistic images. However, ethical concerns, such as copyright protection and generation of inappropriate content, pose significant challenges for the practical deployment of diffusion models. Recent work has proposed a flurry of watermarking techniques that inject visually noteless patterns into generated images, offering a promising solution to these issues. While effective, the essential elements for watermarking and the interconnections among various methods are still chaos. In this paper, we dissect the design principles of state-of-the-art watermarking techniques and introduce a unified framework. We identify a set of dimensions that explain the manipulation enforced by watermarking methods, including the distribution of individual elements, the specification of watermark regions within each channel, and the choice of channels for watermark embedding. Moreover, under this framework we instantiate a new watermarking method to minimize impacts on the model performance from a distributional perspective. Through the empirical studies on regular text-to-image applications and the first systematic attempt on watermarking image-to-image diffusion models, we thoroughly verify the effectiveness of our proposed framework through comprehensive evaluations. On all the diffusion models, including Stable Diffusion, our approach induced from the proposed framework not only preserves image quality but also outperforms existing methods in robustness against a range of attacks.

## 1 Introduction

The rise of diffusion models has significantly impacted image generation, enabling the creation of diverse and high-quality images across various styles. However, the widespread use of these models also introduces critical ethical challenges, particularly concerning copyright protection and the generation of inappropriate or misleading content. In this regard, *watermarking* generated images offers a promising approach to tracing image origins and mitigating potential misuse.

A core challenge in image watermarking is the trade-off between the robustness of the watermark and the quality of the generated images. Traditional watermarking techniques (Al-Haj, 2007; Navas et al., 2008) primarily rely on post-processing methods to embed subtle modifications into the image's frequency representation, making them imperceptible to human users. Though effective, these approaches suffer from reduced image quality and insufficient robustness against common attacks such as compression and cropping, to name a few.

To improve the trade-off, Zhang et al. (2019); Zhu et al. (2018); Hayes & Danezis (2017) proposed end-to-end deep learning methods to construct watermarks with powerful learning capabilities, while such methods are black-box and require additional training. Specifically for diffusion models, Wen et al. (2023); Yang et al. (2024) developed latent-representation-based watermarks, which manipulate latent representations to match specific patterns.

Despite the empirical successes of image watermarking, the key components thereof are poorly understood, and the connections between watermarking methods remain unclear. In this paper, we carefully analyze the state-of-the-art watermarking methods, and introduce a unified framework that identifies and connects the underlying design principles. This framework reveals three critical design dimensions, including ❶ the distribution of individual elements, ❷ the specification of watermark

regions within each channel, and ❸ the choice of channels for embedding. Under the unified framework, we integrate the key designs from both image and language watermarking; we propose a novel, training-free watermarking approach that applies directly to diffusion models without altering the training process. Notably, our method embeds watermarks directly into the latent space of diffusion models (rather than the frequency domain), avoiding extraneous operations and errors during detection. Extensive experiments are conducted; in addition to the conventional focus on text-to-image generation, we extend the application scope of our proposed image watermarking to the scenarios involving image-to-image diffusion model. Our method turns out both preserving visual quality of generated images and providing robustness against a wide range of adversarial attacks.

In summary, our paper contributes the following:

- We present a unified framework that identifies and connects the key design principles underlying state-of-the-art watermarking methods.
- Under the unified framework, we propose a novel, training-free watermarking method, and theoretical analysis showing preservation of the latent representation distribution is accompanied.
- We extend watermarking techniques beyond the traditional focus on text-to-image generation scenario, and give the first systematic approach to watermarking image-to-image diffusion models. Our proposed method exhibits robustness against a variety of adversarial attacks and high visual quality, validating the efficacy of our method as well as the unified framework.

## 2 RELATED WORKS

In this section, we comprehensively review the related works on watermarking images and recent extensions to diffusion models.

**Watermarking Image.** Digital watermarking (Van Schyndel et al., 1994) embeds traceable identification information in carrier data for copyright protection and content authentication. Traditional image watermarking techniques, often applied in post-processing, focus on frequency domain methods (Cox et al., 2007; Al-Haj, 2007; Hamidi et al., 2018; Kundur & Hatzinakos, 1997; Lee et al., 2007; Navas et al., 2008) to enhance robustness. For instance, DwtDctSvd (Cox et al., 2007) combines Discrete Wavelet Transform, Discrete Cosine Transform, and Singular Value Decomposition for watermark embedding. More recently, deep learning-based approaches (Zhu et al., 2018; Tancik et al., 2020; Fernandez et al., 2022; Zhang et al., 2019; Hayes & Danezis, 2017) like RivaGAN (Zhang et al., 2019) leverage neural networks to improve watermarking, employing adversarial networks for both embedding and extraction. Despite advancements, post-hoc watermarking often introduces visible noise and is vulnerable to attacks like cropping and compression (Fernandez et al., 2023), as it is applied to the final image, making the watermark prone to distortion or removal.

**Watermarking Diffusion Models.** With the rise of generative models, particularly diffusion models, watermarking AI-generated content or the models themselves has gained importance. Yu et al. (2021) and Zhao et al. (2023b) proposed embedding watermarks into training datasets, so models inherently generate watermarked content. However, this is impractical for large-scale diffusion models trained on vast datasets. To overcome this, researchers have embedded watermarks during the generation process. For example, Fernandez et al. (2023) and Feng et al. (2024) fine-tuned model weights to modify latent representations, with Stable Signature (Fernandez et al., 2023) fine-tuning LDM decoders to embed hidden watermarks in generated images.

Other approaches, like Tree-ring watermark (Wen et al., 2023), modify the initial noise in the sampling process, requiring deterministic samplers like DDIM (Song et al., 2020) for watermark extraction through inversion. Similarly, Yang et al. (2024) adjusts the initial noise naturally, improving image quality and robustness against attacks. Inspired by these works, we examine critical design dimensions for embedding watermarks in the latent space, offering insights into diffusion watermarking techniques

## 3 PRELIMINARIES

This section recaps key concepts of Latent Diffusion Models (LDMs), focusing on the diffusion process, denoising methods like DDIM, and inversion techniques for watermark detection. We

also introduce image watermarking from a cryptographic perspective, highlighting scenarios for protecting model ownership and detecting illicit image use.

### 3.1 RECAP OF THE DIFFUSION PROCESS

To map a regular image $x \in \mathbb{R}^{H \times W \times 3}$ to the latent space, Latent Diffusion Models (LDMs) formally behave as an autoencoder, using an encoder $E$ to obtain the representation $z_0$ as $z_0 = E(x) \in \mathbb{R}^{h \times w \times c}$; conversely, a decoder $D$ reconstructs the image $x$ from the latent space as $x = D(z_0)$. In generating images, (dropping the encoder $E$ and) directly feeding a random signal $z_0$ to the decoder $D$ of a *pre-trained* LDM can provably return an image following the pre-training distribution.

Despite the autoencoder framework, the magic of LDMs comes from its diffusion-like process to obtain $z_0$. Specifically, an initial latent representation $z_T \in \mathbb{R}^{h \times w \times c}$ is first sampled from a standard Gaussian distribution $\mathcal{N}(0, I)$; subsequently, iterative denoising methods like DDIM (Song et al., 2020) are used to transform $z_T$ into $z_0$, and the decoder then generates the image: $x = D(z_0)$.

Beyond standard LDMs, **inversion techniques** enable moving in the opposite direction—from a generated image back to the initial noise state. It is based on the assumption that the Ordinary Differential Equation (ODE) process can be reversed in the limit of small steps. Dhariwal & Nichol (2021) gave an inverse process based on DDIM (named DDIM inversion), which reconstructs the initial noise tensor $z_T$ given an image $x$. The inversion can be expressed as:

$$\hat{z}_{t+1} = \sqrt{\frac{\alpha_{t+1}}{\alpha_t}} \hat{z}_t + \left( \sqrt{\frac{1}{\alpha_{t+1}} - 1} - \sqrt{\frac{1}{\alpha_t} - 1} \right) \cdot \epsilon_\theta(\hat{z}_t, c), \quad t = 0, 1, \cdots, T-1,$$

where $\alpha_t$ and $\alpha_{t+1}$ are model hyperparameters that control the diffusion process, $\epsilon_\theta(\hat{z}_t, c)$ represents the noise predicted by the model at step $t$, $c$ denotes the conditioning input (such as text prompts or images), and $\hat{z}_0 = z_0 = E(x)$ is the encoded representation of the input image. Previous empirical findings (Wen et al., 2023) suggest that DDIM inversion reliably reconstructs the initial noise, with $\hat{z}_T \approx z_T$. This reliable inversion performance holds for both unconditional and conditional diffusion models, even when the conditioning $c$ is absent.

DDIM inversion can thus be used for watermark detection. Given a generated image $x$ and its associated starting noise $z_T$, we apply DDIM inversion to recover $\hat{z}_T$. This property allows us to compare the reconstructed noise to the original to detect embedded watermarks effectively.

### 3.2 IMAGE WATERMARKING: A CRYPTOLOGICAL PERSPECTIVE

In general, image watermarking strategy is designed for a comprehensive scenario involving three key players: John, the thief Emma, and two types of users, David and Sarah. John is responsible for training the model, deploying it on a platform, and providing an API for users in the case of not open-sourcing the code or model weights. Emma, instead of using John's services, steals images generated by his model and *falsely claims copyright ownership* as if she is the artist. Meanwhile, David and Sarah, as community users, utilize the API to generate and share images. David adheres to community guidelines, while Sarah engages in the creation of *deep fakes and infringing content*.

To evade detection and traceability, Sarah applies various data augmentation techniques to alter illicit images. To prevent this misuse, John embeds a watermark into each generated image. Extracting this watermark not only provides proof of John's rightful ownership but also confirms that the image is artificially generated, distinguishing it from natural images.

## 4 METHODOLOGY

In this section, we will provide a detailed description of each dimension of our watermark. Our goal is to provide a comprehensive guide that details the step-by-step process for watermarking diffusion models, thereby facilitating knowledge transfer and enabling others to replicate and enhance our work.

### 4.1 A Comprehensive Framework for Latent Representation Watermarking

To embed a robust and stealthy watermark within diffusion models, it is crucial to preserve the distribution of the initial latent representation $z_T \in \mathbb{R}^{h \times w \times c}$ (which is intricately linked with image semantics), enhancing detectability against various attacks. To this end, various watermarking methods are raised.

We attempt to categorize watermarking methods for latent representations along three dimensions, which serves as our method framework: ❶ the distribution of individual elements, ❷ the specification of watermark regions within each channel, and ❸ the choice of channels for watermark embedding. The three dimensions constitute a comprehensive image watermarking framework, and each of them all plays a pivotal role in the effectiveness of the watermarking method, impacting both the quality of the generated images and the resilience of the watermark. As a prologue, we decompose existing image watermarking methods under our proposed framework in the following.

**Distribution of Individual Elements.** The first dimension concerns the domain used and how the values are set for each element. Previous work such as Tree-ring (Wen et al., 2023) set the value to a constant in the Fourier frequency domain within each circle, creating a ring-like pattern. Ring-ID (Ci et al., 2024) advances this approach by imprinting the tree-ring pattern only in the real part of the Fourier transform and leaving the imaginary part blank, satisfying the odd function constraint and improving imperceptibility. Post-hoc processing methods like DwtDctSVD (Cox et al., 2007) embed the watermark by modifying wavelet coefficients; they alter the singular values of the Discrete Cosine Transform (DCT) coefficients in the Discrete Wavelet Transform (DWT) domain to encode the watermark. Gaussian-shading (Yang et al., 2024) imprints the watermark in the spatial domain by selecting the value of each element from a sub-distribution of the original, maintaining the overall distribution while embedding the watermark. Notably, learning-based methods, such as Stable Signature (Fernandez et al., 2023) and AquaLoRA (Feng et al., 2024), tuned the model weights to learn the values to inject, allowing the model to embed the watermark seamlessly during the generation process.

**Specification of Watermark Regions.** The second dimension addresses the selection of watermark regions within each channel of size $h \times w$. Tree-ring (Wen et al., 2023) and Ring-ID (Ci et al., 2024) introduce a watermark as a ring shape consisting of several concentric circles in the frequency domain, targeting specific frequency components to achieve a balance between robustness and invisibility. DwtDctSVD (Cox et al., 2007) embeds the watermark into pixel blocks within the combined DWT and DCT frequency domains. Similarly, Gaussian-shading (Yang et al., 2024) defines blocks as the regions to inject the watermark. Learning-based methods typically inject the watermark into all locations within the channel, allowing the model to determine the optimal embedding regions during training.

**Choice of Channels for Watermark Embedding.** The third dimension pertains to the choice of channels for watermark embedding among the $c$ channels. Tree-ring (Wen et al., 2023) and post-hoc processing methods like DwtDctSVD (Cox et al., 2007) added the watermarks into specific channels based on empirical observations, often selecting channels that are less sensitive to perceptual changes to maintain image quality. In contrast, Gaussian-shading (Yang et al., 2024), Ring-ID (Ci et al., 2024), and learning-based methods injected the watermark into all channels, leveraging the entire latent space for embedding and potentially increasing robustness against attacks.

### 4.2 Generalizing Red/Green List for Sampling

As discussed in Section 4.1, previous methods like Tree-ring, Ring-id, and DwtDctSvd heavily relied on setting selected watermarked elements to pre-defined constants. However, the fixed-value operations definitely introduce noticeable artifacts that degrade generation quality, considering the $h \times w \times c$ elements in a latent representation are indeed independent and identically distributed (IID) standard univariate Gaussians $\mathcal{N}(0, 1)$. Moreover, under various attacks these elements can easily be altered, making the watermark difficult to detect.

To overcome these limitations, we propose a distribution-preserving watermarking technique (the characteristic of distribution preservation is reflected in Lemma 4.1). Inspired by the Red/Green

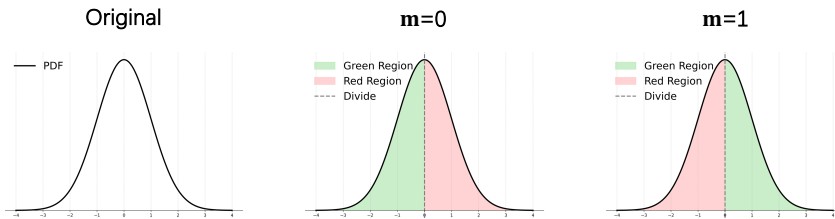

Figure 1: Standard normal distribution partitioning based on watermark value $m$. The left shows the original PDF, while the middle and right plots illustrate the "green" and the "red" domains for $m = 0$ and $m = 1$, respectively, with the dividing line at $x = 0$. Best viewed when zoomed in.

List watermarking method (Kirchenbauer et al., 2023; Zhao et al., 2024b) widely studied in large language models (LLMs), we divide the density function $\phi(x)$ of the standard Gaussian distribution into two domains of equal probability $0.5$. Specifically, we partition the distribution at the origin, creating two truncated distributions disjointly over $(-\infty, 0]$ and $(0, \infty)$. The watermarking process assigns each latent element $z_T^e$ to one of these intervals based on the watermark value $m$, which can be either 0 or 1. This ensures that the latent representations conform to the conditional distribution:

$$p(z_T^e \mid m = i) = \begin{cases} 2 \cdot \phi(z_T^e) & \text{if } Q\left(\frac{i}{2}\right) < z_T^e \leq Q\left(\frac{i+1}{2}\right) \\ 0 & \text{otherwise} \end{cases}$$

where $Q(\cdot)$ is the quantile function of the standard Gaussian distribution $\mathcal{N}(0, 1)$, i.e., the inverse of its cumulative distribution function (CDF). By dividing the continuous standard Gaussian distribution into two portions based on the watermark value $m$, we effectively discretize it to define "green" and "red" domains, similar to the green/red list method in LLM watermarking. The following lemma shows our proposed watermarking technique is distribution-preserving (the proof is deferred to Appendix B.1):

**Lemma 4.1** (Marginal Distribution of Elements). *Every element in the latent representation marginally follows the standard normal distribution $\mathcal{N}(0, 1)$.*

As a closing remark, the key difference between the green/red list method and ours is that, rather than globally sharing the green/red list, we assign different "green" domains—either $(-\infty, 0]$ or $(0, \infty)$—for different elements. Figure 1 illustrates the sampling process of our proposed method. This approach can also be extended to multiple (i.e., more than 2) cumulative probability portions. Importantly, when averaged over all possible watermark values, the marginal distribution $p(z_T^e)$ of the watermarked latent tensor remains the same as the original latent representation.

### 4.3 REGION SPECIFICATION

Compared to LLM watermarking, image watermarking needs to specify the regions (the collections of elements) to be separately watermarked; watermarks that span the entire initial input can be overly sensitive to various transformations, making detection vulnerable to noise, distortions, or local perturbations. In this regard, previous works primarily define watermark embedding regions as ring-shaped (Wen et al., 2023), block-shaped (Cox et al., 2007), or learnable values distributed across all locations (Zhang et al., 2019). Specifically, Cox et al. (2007); Yang et al. (2024) introduced block-based structures, which enhance robustness by localizing the watermark to discrete regions, demonstrating resilience against noise addition, particularly global noise. On the other hand, ring-shaped regions offer robustness against geometric transformations such as rotation. In this section, we will illustrate how to specify the regions for the watermark, to further improve the distribution-preserving generalized red/green list method proposed in Section 4.2; the corresponding watermark detection processes are deferred to the "Detection aggregation along channels" paragraph in Section 4.4.

**"Random Gaussian" Watermarking with Redundant and Dispersed Watermarks.**  Our first idea is to introduce redundancy to attain resistance to attacks. Inspired by Vision Transformers (Dosovitskiy et al., 2020), we split the initial input into patches, each carrying an identical watermark matrix $\boldsymbol{W}$ of the same shape as the patch; every element of $\boldsymbol{W}$ is a watermark value $m$ denoting

the "green" domain. This redundancy strengthens robustness, as information from multiple patches can be aggregated during detection; even if some patches are compromised, ensemble methods help reconstruct the watermark, ensuring reliable detection.

However, redundant patterns over block-shaped patches notably introduces another challenge—the substantial degradation of generation quality due to the artificial structuring of the input. To maintain image fidelity while preserving watermark robustness, we move away from the concept of **fixed block-shaped** patches, and instead suggest **randomly coordinated** regions that exhibit more natural distribution (c.f. the illustration in Figure 2 and the technical analysis in Proposition 4.2). Technically, we will uniformly sample a permutation of the representation elements and then adopt the block-shaped specification, which thoroughly **disperse** the elements. This change reduces the visual artifacts commonly associated with rigid patterns and better preserves the generation quality.

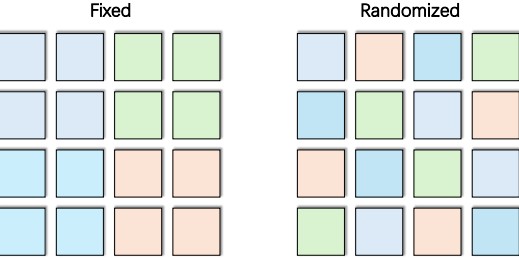

Figure 2: Comparison of Random Gaussian watermarking between fixed and randomized patches. Different colors represent distinct patches. The left plot shows a fixed patch arrangement, while the right plot displays a randomized patch configuration.

To validate the claim that dispersed watermarks induce more natural representation distribution, we show in the following proposition that the correlation of two elements quickly diminish with the patch size $n$, and the limiting covariance matrix is thus akin to the one of a multivariate standard normal distribution. (The proof is provided in Appendix B.2.)

**Proposition 4.2** (Correlation Between Elements). *Let $p$ be the number of patches and $n$ indicate the number of pixels in a patch. The correlation between any two elements $X$ and $Y$ in the tensor is given by*

$$Corr(X, Y) = \frac{2}{\pi} \cdot \frac{p-1}{np-1}.$$

*Furthermore, for a fixed representation size $np$, $Corr(X, Y)$ is exclusively influenced by the number of patches $p$.*

The proposition above suggests selecting a small $p$ to minimize the correlation; however, this choice will damage the robustness, which reflects a trade-off between quality and robustness.

**"Gaussian Ring" Watermarking for Geometric Robustness.** Given the more specific requirement to handle geometric transformations, the preceding approach is too general and thus suboptimal; we introduce an alternative approach to complement our framework: injecting the so-called "Gaussian Rings" into the representation of the latent space.

The concept of "ring" implies a representation tensor is divided by a series of disjoint rings, which naturally inherits the "redundancy" idea and enhances robustness. In detail, each Gaussian Ring is a meticulously structured ring-shaped region carrying a specific watermark value, designed to provide rotational invariance. As shown in Figure 3, the elements on a ring with a specific radius share the same watermark value, all sampled from a truncated Gaussian with a designated ("green") domain.

Unlike previous works that embed rings in the frequency domain (Wen et al., 2023), our method remains entirely within the spatial domain, which simplifies operations and avoids error propagation that can arise from spatial-frequency transformation. Working only in the spatial domain also enables direct manipulation of the latent space without the need for frequency-based transformations, reducing computational overhead and improving integration with mainstream spatial-based generative models.

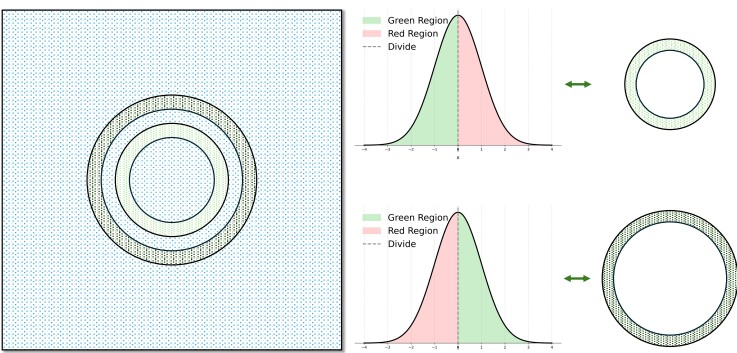

Figure 3: Visualization of Gaussian Ring Watermarking for geometric robustness. The left image shows two Gaussian rings embedded in the latent space, with deep green and light green indicating the two rings. The top and bottom examples on the right illustrate how different truncated Gaussian distribution are mapped to these rings in the latent space.

## 4.4 MULTI-CHANNEL WATERMARKING AND DIRECT APPLICATIONS

We have proposed two types of region specification in Section 4.3. To further enhance the robustness and versatility of our watermarking approach, we propose a hybrid watermarking strategy that combines the strengths of the "Random Gaussian" and "Gaussian Ring" watermarking techniques, through dynamically applying the watermarking techniques according to a third dimension—watermarking channels .

**Imprinting**   To effectively integrate these two methods, we adopt a channel rating strategy to determine the watermarking technique to apply, specifically through calculating the sensitivity to geometric transformations. For a channel $c$ (the notation is slightly abused here), we compute the magnitude of the gradient w.r.t. to a geometric transformation loss $\mathcal{L}_{\text{geo}}$:

$$g_c = \|\partial \mathcal{L}_{\text{geo}}/\partial z_T^c\|_2, \quad \mathcal{L}_{\text{geo}}(z_0, z_0^{\text{rot}}) = \|D(z_0) - D(z_0^{\text{rot}})\|_2^2,$$

where $z_T^c$ represents the latent representation of channel $c$ at time step $T$, $D(z_0), D(z_0^{\text{rot}})$ denote the images generated by $z_T$ before and after 90 degree rotation, respectively, and $\|\cdot\|_2$ denotes the Euclidean norm. Clearly, the channels with larger gradient magnitudes $g_c$ are more sensitive to geometric transformations, and we will apply Gaussian Ring watermarking to enhance their robustness against geometric attacks. Conversely, for channels with smaller gradient magnitudes we will apply Random Gaussian watermarking to better handle non-geometric attacks. In summary, by embedding both random and structured watermark patterns, even if part of the watermarks are compromised by a specific attack, the others can still be detected, enhancing overall robustness.

**Detection aggregation along channels.**   During the watermark detection process, we calculate the accuracy for each channel by evaluating whether the elements of $\hat{z}_T^{(c)}$ (a certain channel of the representation $\hat{z}_T$, an $h \times w$ matrix) fall into the specified "green" region[1]. The so-called "accuracy" $\text{Acc}(z_T^{(c)}, w^c)$ for channel $c$ is calculated as:

$$\text{Acc}(\hat{z}_T^{(c)}, m^c) = \frac{1}{N_c} \sum_{i=1}^{N_c} \mathbf{1}\{z_i^c \in \text{Green Region}(m^c)\}$$

where $N_c$ is the number of watermarked elements in channel $c$, $z_i^c$ is the $i$-th element of channel $c$, $m^c$ is the watermark value for channel $c$, and $\mathbf{1}\{\cdot\}$ is the indicator function, which equals 1 if the event is true and 0 otherwise. The overall watermark recovery accuracy is determined by aggregating the accuracy across all channels. To adaptively address varying attack conditions, we propose the combined watermark accuracy to select the most robust component:

$$\text{Acc}(\hat{m}) = \max_{c \in C_m} \text{Acc}(\hat{z}_T^{(c)}, m^c)$$

---

[1]In addition to the practical detection based on the accuracy metric, a testing procedure is depicted in Appendix B.3.

where $C_m$ is the set of all channels where the watermark is embedded. In practice, we embed the watermark across all channels to enhance overall robustness. The detection process is guided by the intuition that a robust watermark will maintain higher accuracy when compared to ground-truth watermark patterns than those that are not resilient to certain attacks. Our approach allows for the adaptive selection of the most robust watermark under different attack scenarios.

This adaptive approach leverages the advantages of both watermarking techniques, enhancing the overall robustness and reliability of the watermark. Moreover, by combining these methods, the total watermark capacity is greatly increased, as the combined capacity becomes the product of the individual capacities of each technique. The increased capacity allows users to embed more information, improving the efficacy of traceability across diverse use cases. We validate this strategy through empirical studies in Section 5.2.

# 5 EMPIRICAL RESULTS

We conduct experiments on two widely-used diffusion scenarios: text-to-image diffusion models and image-to-image diffusion models, to evaluate the effectiveness and robustness of our watermarking technique across various attack scenarios. Additionally, we perform ablation studies in Section 5.3 for a deeper analysis of the method.

## 5.1 EXPERIMENTAL SETTING

This paper first examines the proposed method on text-to-image diffusion models, with a particular focus on Stable Diffusion (SD) models (Rombach et al., 2022). The generated images have a dimension of $512 \times 512$, while the latent space is sized at $64 \times 64 \times 4$. For inference, we use prompts from Stable-Diffusion-Prompt, setting the guidance scale to 7.5. Image generation is conducted over 50 steps using the DPMSolver (Lu et al., 2022). The watermark radius $r$ ranges from 5 to 15, with an interval of 2. We divide the images into patches, with each patch containing 64 elements. Considering the common user practice of sharing generated images without retaining the original prompts, we perform inversions using an empty prompt and a guidance scale of 1. Inversions are executed with 50 steps using the DDIM method (Song et al., 2020).

We further evaluate the effectiveness of our watermarking approach in image-to-image editing tasks using pre-trained image-conditioned diffusion models. Specifically, we leverage the instruct-pix2pix model (Brooks et al., 2023), a high-performance model fine-tuned from Stable Diffusion on paired image datasets. For these editing tasks, we apply DDIM inversion, again using an empty prompt and an empty original image with a guidance scale of 1. Both inference and inversion are performed with 100 steps to maintain consistency with the original setup, while all other hyperparameters remain consistent with Stable Diffusion.

For evaluating generation quality, we adopt the Frechet Inception Distance (FID) (Heusel et al., 2017) and CLIP-Score (Radford et al., 2021). FID is calculated on the COCO2017 validation set, which contains 5,000 images, to assess image quality. Additionally, prompts from Stable-Diffusion (Rombach et al., 2022) and images from Instructpix2pix (Brooks et al., 2023) are used to generate images. For the Stable Diffusion model, we calculate the CLIP score between the generated images and the text prompts, while for the instruct-pix2pix model, we evaluate the CLIP score between the generated images and the ground truth descriptions.

In the detection scenario, we compute the **true positive rate** (TPR) corresponding to a fixed **false positive rate** (1% FPR). In the traceability scenario, we assess identification accuracy across various watermark patterns. For AUC and TPR@1%FPR, we generate 1,000 watermarked and 1,000 unwatermarked images in each run. All reported metrics are averaged across three runs, each using different random seeds in line with this protocol.

## 5.2 PERFORMANCE OF THE PROPOSED METHOD

To benchmark the robustness of our watermarking method, we document its performance under several widely-used augmentation-based attacks. The details of the attacks are deferred to Appendix A.1. The results, shown in Table 1, present the average detection performance against various types of attacks. These findings demonstrate that our proposed method offers robust watermarking

Table 1: Performance comparison of different watermarking methods on Stable Diffusion under clean and adversarial conditions. The metrics are split into two categories: Fidelity and Robustness.

| Methods | Robustness | | | | | | Fidelity | |
|---|---|---|---|---|---|---|---|---|
| | TPR @1%FPR | | AUC | | Acc | | FID ($\downarrow$) | CLIP-Score |
| | Clean | Adversarial | Clean | Adversarial | Clean | Adversarial | | |
| Stable Diffusion | - | - | - | - | - | - | 25.23 | 0.363 |
| DwtDct | 0.909 | 0.166 | 0.974 | 0.574 | 0.950 | 0.527 | 25.28 | 0.364 |
| DwtDctSvd | 1.000 | 0.269 | 1.000 | 0.702 | 1.000 | 0.691 | 25.01 | 0.359 |
| RivaGAN | 0.997 | 0.472 | 0.999 | 0.854 | 0.998 | 0.802 | **24.51** | 0.361 |
| Stable Signature | 1.000 | 0.414 | 1.000 | 0.818 | 1.000 | 0.751 | 25.45 | **0.364** |
| Tree-Ring | 1.000 | 0.922 | 1.000 | 0.993 | 1.000 | 0.979 | 25.29 | 0.363 |
| Gaussian Shading | 1.000 | 0.809 | 1.000 | 0.911 | 1.000 | 0.864 | 25.20 | 0.364 |
| AquaLoRA | 1.000 | 0.738 | 1.000 | 0.871 | 1.000 | 0.817 | 25.50 | 0.363 |
| **Ours** | **1.000** | **0.984** | **1.000** | **0.999** | **1.000** | **0.994** | 25.20 | 0.363 |

Table 2: TPR@1%FPR under different attacks for Stable Diffusion, showing the effectiveness of our method over a number of attacks.

| Method | Clean | Rotation | JPEG | Cr. & Sc. | Blurring | GauNoise | Color Jitter | S&PNoise | DeNoise | Flip | Avg |
|---|---|---|---|---|---|---|---|---|---|---|---|
| DwtDct | 0.909 | 0.027 | 0.008 | 0.092 | 0.011 | 0.354 | 0.126 | 0.089 | 0.016 | 0.023 | 0.166 |
| DwtDctSvd | 1.000 | 0.011 | 0.156 | 0.057 | 0.538 | 0.732 | 0.117 | 0.021 | 0.018 | 0.042 | 0.269 |
| RivaGan | 0.997 | 0.012 | 0.756 | 0.762 | 0.428 | 0.541 | 0.694 | 0.477 | 0.025 | 0.027 | 0.472 |
| Stable Signature | 1.000 | 0.032 | 0.713 | 0.816 | 0.015 | 0.624 | 0.843 | 0.072 | 0.011 | 0.018 | 0.414 |
| Tree-Ring | 1.000 | 0.477 | 0.995 | 0.932 | 0.999 | 0.926 | 0.900 | 0.987 | 1.000 | 1.000 | 0.922 |
| Gaussian Shading | 1.000 | 0.007 | 0.999 | 1.000 | 1.000 | 0.999 | 0.992 | 0.999 | 1.000 | 0.097 | 0.809 |
| AquaLoRA | 1.000 | 0.013 | 0.987 | 0.941 | 1.000 | 0.954 | 0.847 | 0.693 | 0.812 | 0.133 | 0.738 |
| **Ours** | **1.000** | **0.852** | **1.000** | **1.000** | **1.000** | **0.996** | **0.996** | **1.000** | **1.000** | **0.998** | **0.984** |

solutions, outperforming different baselines in withstanding a wide range of adversarial manipulations. In terms of fidelity, our method achieves comparable performance to other techniques, with similar FID and CLIP-Score values, indicating minimal impact on image quality.

Further, we show the TPR@1%FPR for each attack setting in Table 2. Our spatial domain watermarking method outperforms Tree-Ring, particularly under rotational attacks (0.852 vs. 0.477) and various noisy conditions, such as Gaussian noise (0.996 vs. 0.926). While Gaussian Shading excels in noisy settings (e.g., 1.000 for blurring), it struggles with geometric transformations like rotation (0.007). Overall, our method achieves the highest average performance, demonstrating outstanding robustness across both geometric and noise-based adversarial attacks. More experimental results on the instruct-pix2pix model and the image-to-image edit task are shown in Table 6 in Appendix A.2. The analysis of watermark capacity and identification is shown in Appendix A.3. All the experimental results demonstrate the superior performance of our method.

### 5.3 ABLATION STUDIES

We conduct extensive ablation studies on several key hyperparameters of our proposed method to demonstrate the effectiveness of our method.

To validate the **generalization** of our approach, we evaluated five commonly used sampling methods in diffusion: DDIM (Song et al., 2020), UniPC (Zhao et al., 2024a), PNDM (Liu et al., 2022), DEIS (Zhang & Chen, 2022), and DPMSolver (Lu et al., 2022). As shown in Table 3, with our proposed watermarking technique, all sampling methods demonstrate excellent and comparable performance, particularly in clean conditions where all methods achieved a perfect detection rate. Under adversarial noise, DPMSolver shows a marginally better detection rate, but overall, all sampling methods maintain high robustness.

Table 4 shows that **increasing the patch size** improves robustness against adversarial attacks but reduces image quality, as reflected by a lower CLIP-Score. For the **ring radius**, placing the ring near the center (c.f. the column "0-5" in Table 5) harms generation quality since it embeds key structural and semantic information for diffusion. The medium ring radius ("5-15") offers the best balance, providing strong rotation robustness, while maintaining image quality. Due to space constraints, the ablation study on Gaussian Ring and Random Gaussian watermarks is provided in Appendix A.4,

Table 3: Detection TPR with different sampling methods in diffusion models.

| Noise | Sampling Method | | | | |
|---|---|---|---|---|---|
| | DDIM | UniPC | PNDM | DEIS | DPMSolver |
| Clean | 1.000 | 1.000 | 1.000 | 1.000 | 1.000 |
| Adversarial | 0.978 | 0.972 | 0.983 | 0.982 | 0.984 |
| CLIP-Score | 0.363 | 0.362 | 0.363 | 0.363 | 0.363 |

Table 4: Detection TPR@1%TPR with different patch sizes.

| Patch Size | 4 | 16 | 64 | 256 |
|---|---|---|---|---|
| None | 0.993 | 1.000 | 1.000 | 1.000 |
| Adversarial | 0.757 | 0.924 | 0.984 | 0.986 |
| CLIP-Score | 0.364 | 0.363 | 0.363 | 0.359 |

Table 5: Detection TPR@1%TPR with different ring radii.

| Ring Radius | 0-5 | 5-10 | 0-10 | 5-15 | 10-15 |
|---|---|---|---|---|---|
| None | 1.000 | 1.000 | 1.000 | 1.000 | 1.000 |
| Rotation | 0.512 | 0.643 | 0.828 | 0.852 | 0.767 |
| CLIP-Score | 0.361 | 0.363 | 0.359 | 0.363 | 0.363 |

the analysis of inversion and inference steps is in Appendix A.5, and the impact of channels is discussed in Appendix A.6. All results demonstrate the effectiveness of our method.

### 5.4 VISUALIZATION

In this subsection, we visualize the watermarked and non-watermarked diffusion-generated images in Figure 4. It demonstrates that our watermarked images successfully preserve the semantic information and maintain high visual quality across different scenarios. In text-to-image diffusion, the watermarked images remain visually similar to the non-watermarked images while keeping the intended semantics intact. In the image-to-image diffusion setting, the watermarked images exhibit even greater visual similarity to both the original and non-watermarked images, due to the input image providing additional guidance. These illustrations empirically validate that our watermarking method effectively maintains image integrity while embedding the watermark.

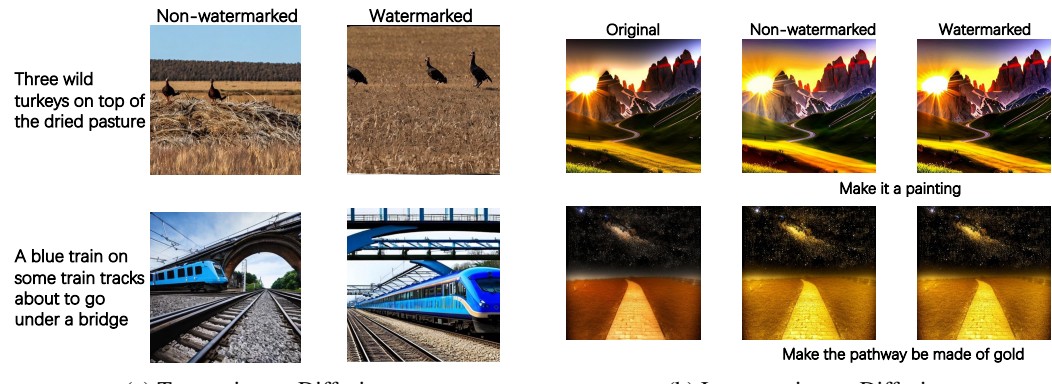

(a) Text-to-image Diffusion        (b) Image-to-image Diffusion

Figure 4: Visualization of both the watermarked and non-watermarked generated images in different scenarios. For image-to-image editing, we also include the original images.

### 6 CONCLUDING REMARKS

In this paper, we introduce a unified framework to dissect watermarking approaches for diffusion models along three distinct dimensions. By further adapting the principle of LLM watermarking, we innovatively instantiate a more effective hybrid model under this framework, maintaining high fidelity, ensuring robust watermarking against a variety of attacks, and attaining minimal computational overhead. We extensively evaluate the resulting model derived from the framework on text-to-image applications, which outperforms existing state-of-the-art image watermarking methods. Moreover, we directly apply the proposed model to less studied image-to-image diffusion models; the exceptional performance further evidence a significant advancement in digital watermarking.

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

# A   MORE ON EXPERIMENTS

The code and model weights will be open-sourced after the review procedure.

## A.1   IMPLEMENTATION DETAILS

The attacks implemented in the paper include Gaussian blurring with a filter size of 4, Gaussian noise with a standard deviation of 0.05, JPEG compression with a ratio of 25%, and salt-and-pepper noise with a probability of 0.05. Additionally, we evaluate against brightness adjustments where the factor is set to 6, random cropping and resizing with a 75% ratio, rotation by 75 degrees, and horizontal and vertical flips with probabilities of 0.5 each. We also incorporate a model-based attack, DeNoise, as introduced by Zhao et al. (2023a), which leverages a diffusion model to denoise the output, aiming to remove any potential watermark.

For the Gaussian Ring Watermark, the watermark radius $r$ ranges from 5 to 15, with an interval of 2. For the Random Gaussian Watermark, we divide the images into patches, with each patch containing 64 elements. Furthermore, to enhance the randomness and security of the watermark, we employ a stream key—a cryptographic key used in encryption algorithms to generate a sequence of pseudo-random values. Following Yang et al. (2024), we employ a stream key to encrypt the watermark, which consists of binary values (0/1), into a randomized version $\mathbf{m}$ using an encryption method like ChaCha20 (Bernstein et al., 2008). The encrypted watermark $W$, now uniformly distributed, avoids detectable artifacts while maintaining alignment with the natural data distribution.

## A.2   EMPIRICAL RESULTS ON INSTRUCT-PIX2PIX

Table 6: AUC under each Attack for Instruct-pix2pix image-to-image diffusion, showing the effectiveness of our method over a number of augmentations.

| Method | Clean | Rotation | JPEG | Cr. & Sc. | Blurring | GauNoise | Color Jitter | S&PNoise | DeNoise | Flip | Avg |
|---|---|---|---|---|---|---|---|---|---|---|---|
| Tree-Ring | 0.751 | 0.586 | 0.689 | 0.653 | 0.695 | 0.625 | 0.615 | 0.673 | 0.602 | 0.605 | 0.649 |
| Gaussian Shading | 0.963 | 0.532 | 0.961 | 0.924 | 0.943 | 0.912 | 0.942 | 0.912 | 0.916 | 0.525 | 0.853 |
| **Ours** | **0.988** | **0.830** | **0.976** | **0.947** | **0.970** | **0.914** | **0.946** | **0.920** | **0.942** | **0.841** | **0.927** |

The experimental findings in Table 6 demonstrate that, even in the more challenging image-to-image diffusion scenarios, our method maintains high robustness across various attack types, with an average AUC of 0.927. In contrast, the performance of the Tree-Ring method significantly declines, possibly because it sets selected values to constants, heavily relying on inversion precision. In image-to-image settings, inversion is more difficult due to the absence of multiple inputs including the conditional editing prompt and input image during the inversion process. This highlights the strength of our method, which can achieve robustness even when inversion accuracy is compromised, further underscoring its effectiveness in complex scenarios.

## A.3   WATERMARK CAPACITY AND IDENTIFICATION

Table 7: Traceability and identification accuracy across 32 distinct watermark patterns.

| Method | Clean | Rotation | JPEG | Cr. & Sc. | Color Jitter | GauNoise | Avg |
|---|---|---|---|---|---|---|---|
| Tree-Ring | 0.435 | 0.012 | 0.401 | 0.045 | 0.412 | 0.505 | 0.302 |
| **Ours** | **1.000** | **0.828** | **0.992** | **0.984** | **0.980** | **0.982** | **0.961** |

The experimental results in Table 7 demonstrate the superior capacity and robustness of our watermarking method compared to the Tree-Ring approach. In our method, two distinct watermarks are injected into different channels, each carrying $a$ and $b$ bits of information. This results in $2^a$ and $2^b$ possible patterns for each channel, and when combined, the total capacity becomes $2^{a+b}$. This significantly increases the capacity and enhances the distinguishability of the watermarks compared to constant value-based approaches like Tree-Ring. The results show that across various use cases and watermark patterns, our method consistently achieves high identification accuracy, while the Tree-Ring approach fails, especially under adversarial conditions. These findings underscore

the effectiveness of our method in maintaining both traceability and identification accuracy, even in challenging adversarial scenarios.

## A.4 ABLATION STUDY ON RANDOM GAUSSIAN AND GAUSSIAN RING

Table 8: Comparison of detection performance (TPR@1%FPR) across various methods under different attack scenarios, including Random Gaussian removal, Gaussian Ring removal, and the proposed method (Ours).

| Method | Clean | Rotation | JPEG | Cr. & Sc. | Blurring | GauNoise | Color Jitter | S&PNoise | DeNoise | Flip | Avg |
|---|---|---|---|---|---|---|---|---|---|---|---|
| w/o Random Gaussian | 1.000 | 0.841 | 0.966 | 0.954 | 0.937 | 0.923 | 0.982 | 0.984 | 0.931 | 0.997 | 0.952 |
| w/o Gaussian Ring | 1.000 | 0.011 | 1.000 | 1.000 | 0.986 | 0.994 | 0.988 | 0.996 | 1.000 | 0.015 | 0.799 |
| **Ours** | **1.000** | **0.852** | **1.000** | **1.000** | **1.000** | **0.996** | **0.996** | **1.000** | **1.000** | **0.998** | **0.984** |

The experimental results in the Table 8 show that both the Random Gaussian and Gaussian Ring watermark patterns are critical to the success of our method. The Random Gaussian watermark demonstrates greater robustness against noise-based attacks, as indicated by the high performance in noise-related tests (e.g., Salt & Pepper Noise: 0.996), while the Gaussian Ring watermark shows superior robustness to geometric transformations, such as rotation (0.011 for without Gaussian Ring, compared to 0.841 with Gaussian Ring). Despite this, both patterns perform well individually, achieving strong results across various attack scenarios. The combination of the two watermarks in our method results in the highest overall performance (average 0.984), demonstrating that these patterns are not only effective but also complementary, enhancing the overall robustness when used together.

## A.5 ABLATION STUDY ON INFERENCE AND INVERSION STEPS.

Table 9: Detection TPR@1%FPR with different inversion and inference steps.

| | Inversion Step | | | |
|---|---|---|---|---|
| **Inference Step** | **10** | **25** | **50** | **100** |
| 10 | 0.975 | 0.976 | 0.973 | 0.970 |
| 25 | 0.968 | 0.978 | 0.981 | 0.981 |
| 50 | 0.965 | 0.967 | 0.984 | 0.982 |
| 100 | 0.965 | 0.966 | 0.977 | 0.984 |

The experimental findings in Table 9 demonstrate that different inversion steps consistently perform well in terms of detection accuracy, with minimal loss even when there is a mismatch between inference and inversion steps. In real-world scenarios, the exact inference step is often unknown, which can result in this mismatch. However, the table shows that detection performance remains robust across various combinations of steps. Given the efficiency of existing samplers and the optimal performance observed with 50 inversion steps, we select 50 steps as a balanced trade-off between accuracy and computational efficiency.

## A.6 ANALYSIS ON CHANNELS

The experimental findings in Figure 5 show substantial variation in performance across different channels, with Channel 2 achieving the highest accuracy. Notably, the computed gradient values align closely with the robustness accuracy for each channel, indicating that gradient strength is a reliable indicator of channel performance. This suggests that channels with stronger gradients are better suited for embedding the Gaussian Ring Watermark, providing a useful guideline for selecting the optimal channel for watermarking to enhance specific robustness.

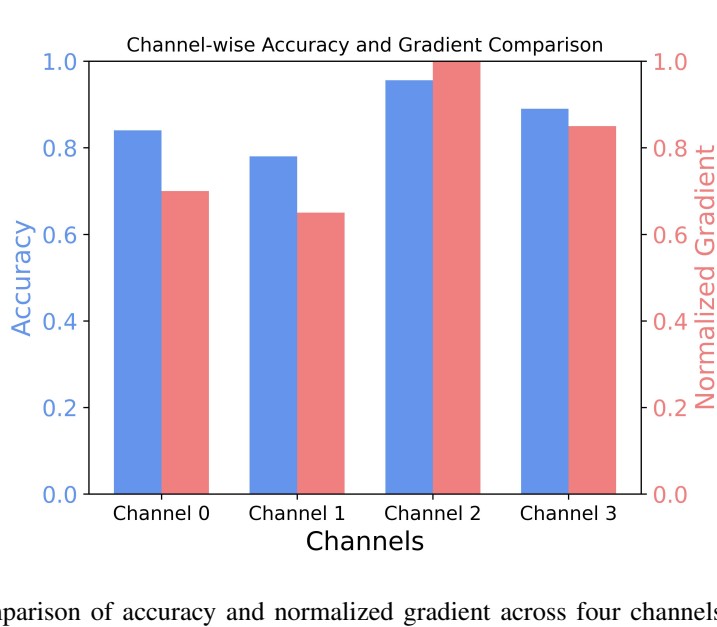

Figure 5: Comparison of accuracy and normalized gradient across four channels. Each channel contains two bars: the blue bar represents the accuracy for that channel, while the red bar shows the corresponding normalized gradient.

# B  PROOFS OMITTED IN THE MAIN TEXT

## B.1  PROOF OF LEMMA 4.1

*Proof.* Let $Z$ be a random variable representing the value of an element in the latent representation. The pixel $Z$ is sampled based on a watermark bit $B \in \{0, 1\}$, which determines the region of the Gaussian distribution from which $Z$ is drawn.

The conditional distribution of $Z$ given $B$ is:

$$p_{Z|B}(z_T^e|w) = \begin{cases} 2\phi(z_T^e), & \text{if } z_T^e \in R(w), \\ 0, & \text{otherwise,} \end{cases}$$

where $\phi(x) = \dfrac{1}{\sqrt{2\pi}}e^{-x^2/2}$ is the standard normal probability density function, and $R(w)$ is defined as:

$$R(w) = \begin{cases} (-\infty, 0], & \text{if } w = 0, \\ (0, \infty), & \text{if } w = 1. \end{cases}$$

The watermark bits $B$ are assumed to be independent and uniformly random, i.e., $P(B = 0) = P(B = 1) = \dfrac{1}{2}$. The marginal distribution of $Z$ is then:

$$\begin{aligned} p_Z(z_T^e) &= \sum_{w \in \{0,1\}} p_{Z|B}(z_T^e|w)P(B = w) \\ &= \frac{1}{2} \cdot 2\phi(z_T^e)\mathbf{1}_{z_T^e \leq 0} + \frac{1}{2} \cdot 2\phi(z_T^e)\mathbf{1}_{z_T^e > 0} \\ &= \phi(z_T^e)\left[\mathbf{1}_{z_T^e \leq 0} + \mathbf{1}_{z_T^e > 0}\right] \\ &= \phi(z_T^e). \end{aligned}$$

Therefore, $Z$ marginally follows the standard normal distribution $\mathcal{N}(0, 1)$. ◇

## B.2 PROOF OF PROPOSITION 4.2

*Proof.* Consider two elements $X$ and $Y$ within a latent representation divided into $p$ patches, each containing $n$ pixels, totaling $N = pn$ pixels. Pixels within the same patch share a common watermark bit $W$, while those in different patches have independent watermark bits.

$$\text{Cov}(X, Y) = \mathbb{E}[XY] - \mathbb{E}[X]\mathbb{E}[Y].$$

Given $\mathbb{E}[X] = \mathbb{E}[Y] = 0$, this simplifies to:

$$\text{Cov}(X, Y) = \mathbb{E}[XY].$$

When $X$ and $Y$ are in the same position across different patches, they share the same watermark bit $W$. Given $W$, $X$ and $Y$ are independent and follow a Half-Normal Distribution with variance $\sigma^2 = 1$.

The probability density function (PDF) of a Half-Normal Distribution with variance $\sigma^2 = 1$ is

$$f_Y(y) = \sqrt{\frac{2}{\pi}} \exp\left(-\frac{y^2}{2}\right), \quad y \geq 0.$$

The expectation $\mathbb{E}[Y]$ is calculated as

$$
\begin{aligned}
\mathbb{E}[Y|W=1] &= \int_0^\infty y \cdot f_Y(y) \, dy \\
&= \sqrt{\frac{2}{\pi}} \int_0^\infty \sqrt{2t} \cdot e^{-t} \cdot \frac{1}{\sqrt{2t}} \, dt \quad (\text{where } t = \frac{y^2}{2}) \\
&= \sqrt{\frac{2}{\pi}}.
\end{aligned}
$$

Similarly, for $W = 0$,

$$\mathbb{E}[Y|W=0] = -\sqrt{\frac{2}{\pi}}.$$

Since $X$ and $Y$ are independent given $W$,

$$\mathbb{E}[XY|W] = \mathbb{E}[X|W]\mathbb{E}[Y|W] = \left(\sqrt{\frac{2}{\pi}}\right)^2 = \frac{2}{\pi}.$$

When $X$ and $Y$ are in different positions across different patches, their watermark bits $W_X$ and $W_Y$ are independent. Therefore,

$$\mathbb{E}[XY|\text{Different Positions}] = \mathbb{E}[X]\mathbb{E}[Y] = 0.$$

The probability that two randomly selected pixels are in the same position across different patches is

$$P_s = \frac{\#\text{Same Position}}{\#\text{Total}} = \frac{p-1}{np-1}$$

Combining the cases, we have

$$\mathbb{E}[XY] = P_s \cdot \frac{2}{\pi} + (1 - P_s) \cdot 0 = \frac{2}{\pi} \cdot \frac{p-1}{np-1}.$$

Given that $\text{Var}(X) = \text{Var}(Y) = 1$,

$$\text{Corr}(X, Y) = \frac{\text{Cov}(X, Y)}{\sqrt{\text{Var}(X)\text{Var}(Y)}} = \frac{2}{\pi} \cdot \frac{p-1}{np-1}.$$

$\Diamond$

## B.3 ILLUSTRATION OF THE STATISTICAL TEST

In this work, our primary focus is on evaluating the actual performance of the watermarking method. However, a statistical analysis can also be derived. Let $m \in \{0, 1\}^k$ represent a $k$-bit (independent) watermark embedded in the model. We extract the message $m'$ from an image $x$ and compare it with $m$. As outlined in previous works, the detection test is based on the number of matching bits, $A(m, m')$. Specifically, if

$$A(m, m') \geq \tau \quad \text{where} \quad \tau \in \{0, \ldots, k\},$$

then the image is flagged. This approach provides a level of robustness against imperfections in the watermarking process.

Formally, we test the statistical hypothesis $H_1$: "image $x$ was generated by the watermarked model" against the null hypothesis $H_0$: "image $x$ was not generated by the watermarked model." Under $H_0$ (i.e., for non-watermarked images), we assume that the bits $m'_1, \ldots, m'_k$ are independent and identically distributed (i.i.d.) Bernoulli random variables with a parameter of $0.5$. Consequently, $A(m, m')$ follows a binomial distribution with parameters $(k, 0.5)$. This assumption has been experimentally validated.

The theoretical FPR is defined as the probability that $A(m, m')$ exceeds the threshold $\tau$. It is calculated using the CDF of the binomial distribution. A closed-form expression can be derived using the regularized incomplete beta function $I_x(\alpha; \beta)$:

$$\text{FPR}(\tau) = \mathbb{P}(M > \tau | H_0) = I_{1/2}(\tau + 1, k - \tau).$$

