# OpenReview forum: "Robust Watermarking for Diffusion Models: A Unified Multi-Dimensional Recipe"
_ICLR.cc/2025/Conference — ICLR 2025 Conference Withdrawn Submission_

### Official Review · Reviewer_4EPS · 2024-10-30

**Soundness:** 1
**Presentation:** 1
**Contribution:** 1
**Rating:** 1
**Confidence:** 3

**Summary:**

The authors first summarized and discussed existing watermarking approaches. They identified some common properties in the existing watermarking approaches. The authors proposed three techniques related generalizing red/green list for sampling, region specification and multi-channel watermarking and provided some theoretical results to justify their methods.

**Strengths:**

1)	It outperforms the baseline, in terms of robustness
2)	The fidelity scores, including FID and CLIP-Score are similar to other methods

**Weaknesses:**

1)	The presentation is low than the standard. I need a lot of effort to figure out the algorithm details. In fact, the algorithm is not very complex. The authors need to rewrite entire the algorithm section.
2)	It contains unnecessary information, e.g.,  section 3.2 image watermarking: a cryptological perspective.
3)	Section 4.1, under methodology, it is in fact a short literature review. To me, it is not a framework proposed by the authors. It is just a summary of the existing work. When discussing the related works, better to separate tradition watermarking methods and watermarking methods for generative models.
4)	The proposed method has no way to control the generation quality.
5)	The repeated patches do not improve the robustness but in fact waken it. Let X=[x1… x9] be a 9-dimenional random variable and Pr(xi=0)=0.3 and Pro(xi=1)=0.7.  Let us group x1-x3 be the first group, x4-x6 be the second group and the third group. We use majority voting to determine the group label, 1 or 0. If I am not wrong, Pro(at least 2/3 group label being 1) should be smaller than Pro(at least 2/3 original xi being 1). The authors do not provide any evidence to justify that it is in fact better than not using repeated patch.
6)	The lemma 4.1 is meaningless to me after I read the proof. The lemma in 4.1 causes more confusion. Line 235, the authors use a complex way to present the condition Q(i/2) and Q(i+1/2). I am not sure whether they intend to make the presentation as complex as possible.
7)	The decoding process is not clear. I assume that they use inverse process to get back the noise.
8)	Only several images are provided for visual comparison. They are likely clearly selected. The authors selected incorrectly generated images, but the watermarked scheme correct the mistake (Fig. 4a). What we are interested in is that how many times watermarking scheme provide correct results, but the original generation provides wrong and vice versa.
9)	The evaluation does not include GAI based editing.

**Questions:**

See the weaknesses

---

### Official Review · Reviewer_Cvpo · 2024-11-02

**Soundness:** 3
**Presentation:** 3
**Contribution:** 2
**Rating:** 5
**Confidence:** 3

**Summary:**

The paper proposed a robust watermarking method that can embed watermarks in a unified fashion including text-to-image watermarking and image-to-image watermarking. It identifies three critical dimensions of recent diffusion model-based watermarking methods, discusses three dimensions in depth, and proposes optimized methods for each. For the dimension of distribution of individual elements, the paper uses red/green list watermarking method [1, 2] that is widely studied in LLM. The paper emphasizes the importance of choosing watermarking regions, and proposed two methods, one that uses redundant and dispersed watermarks, the other watermarks with a ring-like shape. To further enhance the robustness and versatility, The paper then proposed a dynamic way to dynamically choose channels and regions. In the experiment, the paper demonstrates strong robustness, while the image quality is not negatively affected too much.

[1] John Kirchenbauer, Jonas Geiping, Yuxin Wen, Jonathan Katz, Ian Miers, and Tom Goldstein. A
watermark for large language models. In International Conference on Machine Learning, pp.
17061–17084. PMLR, 2023.

[2] Xuandong Zhao, Prabhanjan Vijendra Ananth, Lei Li, and Yu-Xiang Wang. Provable robust water-
marking for ai-generated text. In The Twelfth International Conference on Learning Representa-
tions, 2024b.

**Strengths:**

1. The experiment shows solid results. The robustness is state-of-the-art, compared with all the baselines. And this method doesn’t affect the image quality too much.
2. The paper proposed a unified way to embed watermarks, including text-to-image and image-to-image. The text-to-image result shows visual similarity between watermarked image and non-watermarked image, and the image-to-image result shows consistency between the original one and the watermarked one.
3. The paper clearly dissects three dimensions that is critical and common in recent image watermarking method about diffusion models. The proposed method is clearly stated.

**Weaknesses:**

1. The paper doesn’t quantify the consistency between the watermarked image and the non-watermarked image in the case of image-to-image watermarking. The authors are encouraged to use metrics like PSNR, SSIM and LPIPS.
2. While the paper claims that the dynamic watermarking technique increases robustness, it is primarily due to selecting the maximum score among all options. My concern is that different watermark patterns have bias towards different attacks, which lead to inaccuracy in presenting the performance of your method. (For example, for the attack of rotation, the gaussian ring is used, because it's better than that of random gaussian. But for other attacks random gaussian might be better, rather than gaussian ring.) The authors didn’t show readers this point. The authors are recommended to demonstrate TPR@1%FPR score across all attackers, but this time only use one watermarking method, like only use ‘random gaussian’ for all the images.
3. The paper proposed embedding the watermark on the spatial domain, rather than the frequency domain. I have concerns about this. Embedding watermarks in the spatial domain may result in a concentrated watermark information in a specific area. For example if a circle-like watermark is embedded in the central area of latent z_T, then if the central area of an image x_0 is dropped, the watermark may no longer be detectable.

**Questions:**

1. On line 343, the author mentioned that selecting a good watermarking channel can enhance the robustness and versatility. I don’t know the specific meaning of ‘versatility’ in the sentence. Can the authors elaborate on it?
2. I want to discuss my broader concerns regarding this paper and other similar papers that adding watermarks in the latent noise z_T. I appreciate the authors’ significant efforts in coming up with an optimized method regarding three dimensions of the watermarking process. But, it seems they *focused too much* on the latent $z_T$, rather than the diffusion model. This gives me a feeling that this paper is "stacking methods". In the paper, the diffusion model is simply being treated as an encoder-decoder structure connecting the latent z and the image x. If diffusion model is serve in this way, could it be possible that this method can be applied to other similar models like VAE or GANs? Additionally, if time permits, the author can demonstrate some diffusion model-specific components/properties that drives them to only investigate diffusion model exclusively as the model in the experiments.


The score is negotiable. Would like to hear feedback from authors.

---

### Official Review · Reviewer_GShn · 2024-11-03

**Soundness:** 2
**Presentation:** 3
**Contribution:** 2
**Rating:** 5
**Confidence:** 5

**Summary:**

This paper focuses on the robust watermarking of generated image by Diffusion Models. To this end, the paper summarizes the framework of previous methods to deal with such problems, which consists of three main parts: the distribution of individual elements, the specification of watermark regions, and the choice of channels for embedding watermarks for watermark embedding. Based on the analysis of existing methods, this paper tries to optimize each component of the framework.

**Strengths:**

1. This paper summarizes a framework based on the previous method, which has a clear structure and is easy to understand and apply.
2. Compared with other existing methods, the proposed optimization measures improve the robustness of watermarking against geometric and non-geometric attacks

**Weaknesses:**

1. The method has certain limitations in terms of innovation, mainly due to the fact that it constructs a framework by summarizing the ideas of previous methods and then makes some improvements based on previous methods.

2. Furthermore, regarding the three parts improved within the framework, their specific contributions to the overall performance enhancement have not been clearly identified. This has led to the improvement advantages of the method being unclear.

3.The ablation study results provided in the main text also seem inadequate, failing to sufficiently demonstrate the advantages of the proposed method. Although some parameters have been explored, more critical experimental data have been placed in the appendix, which might impact the reader's comprehensive understanding of the method's effectiveness.

**Questions:**

1、Why should a Gaussian distribution be followed during the Distribution of Individual Elements phase, and what are its advantages?
2、This paper adjust the selection method of the watermark area (random Gaussian and Gaussian-ring) by introducing the parameter. However, in comparative experiments with these two methods, it was found that the proposed method’s performance is similar to the better-performing one among them.
3、Regarding the three parts improved within the framework, their specific contributions to the overall performance enhancement have not been clearly identified. Therefore, it is recommended to discuss the specific roles and extent of each part in improving watermark robustness, so as to better demonstrate the effects of the improvements.
4、To strengthen the persuasiveness of the paper, it is suggested that the most critical experimental results that best highlight the method’s advantages be directly presented in the main text, ensuring that readers can easily understand the actual improvements of the proposed method compared to existing technologies.

---

### Official Review · Reviewer_GQNM · 2024-11-04

**Soundness:** 3
**Presentation:** 3
**Contribution:** 3
**Rating:** 6
**Confidence:** 3

**Summary:**

This paper analyzes and summarizes the design principles of existing watermarking methods for the content generated by diffusion models, and proposes a new unified method based on the analysis of the manipulation for watermarking, concerning three dimensions: distribution of individual elements, specification of watermark regions, and choice of channels.

**Strengths:**

1. The proposed watermarking method outperforms the competitive methods, in terms of both robustness and fidelity.
2. The analysis of the three dimensions of the manipulation for watermarking is insightful.
3. The paper is well-written.

**Weaknesses:**

1. The quantitative evaluation of image fidelity does not consider individual-level fidelity. Although the visualization of the individual samples of watermarked and non-watermarked images is shown, the visual similarity is not computed quantitatively.
2. The evaluation of robustness mainly considers the attacks related to geometric transformation, blurring, and additive perturbation. However, in practice, the generated image may go through attacks like image-to-image translation or image editing. The robustness against such attacks is not evaluated in this paper.

**Questions:**

1. Since the Gaussian ring patterns are introduced to the latent space, are there any ring patterns in the pixel space? If there are, are they detectable in other spaces such as the frequency domain?
2. The paper only compares the Random Gaussian watermarking between fixed and randomized patch arrangements. Is it possible that there are some better arrangements, for example, the checkerboard-like patch arrangement?

---

### Official Review · Reviewer_Cq4V · 2024-11-04

**Soundness:** 2
**Presentation:** 2
**Contribution:** 2
**Rating:** 5
**Confidence:** 4

**Summary:**

This paper introduces a framework for optimizing watermarking in diffusion models by thoroughly analyzing key dimensions such as element distribution and embedding channels.

**Strengths:**

* Proposing a new watermarking strategy to improve the robustness and fidelity by systematically analysing the latent-based watermarking methods.

* The idea of "Gaussian rings" is simple yet deliver the promising results as demonstrated in the experimental results.

**Weaknesses:**

* This paper is not well-written, I can only get to the point when it comes to the description of methodology.
* In terms of technical contribution, I suppose it to be limited. The "gaussian rings" may be just slight modifications of exisiting methods mentioned in the related work, e.g. Tree Ring.
* I found several statements can be controlversial. For example: *"However, the fixed-value operations definitely introduce noticeable artifacts that degrade generation quality"* in L211. Are there any experiments to support this? Or any reasonable intuition introduced?

**Questions:**

Please refer to the weaknesses.

---

### Note · Authors · 2024-11-15

**Comment:**

Thanks to the reviewers and Area Chair for their efforts and insightful comments.

---
Moreover, we would like to address a very comment of Reviewer 4EPS. Specifically, we would like to clarify their counterexample in weakness 5 is wrong.


1. **First Scenario: Probability of At Least 2 Out of 3 Group Labels Being Correct**

   For group-level correctness, let $Y \sim \text{Binomial}(3, 0.7)$, representing the correctness of individual group labels. The probability of at least 2 individual labels in a group being correct is: $p = P(Y \geq 2) = P(Y=2) + P(Y=3)$=0.784.
   For the overall group label correctness, let $Z \sim \text{Binomial}(3, p)$, where $p$ is the group-level correctness probability. The probability of at least 2 out of 3 group labels being correct is:

   $$
   P_1 = P(Z \geq 2) = \sum_{k=2}^{3} \binom{3}{k} p^k (1 - p)^{3-k} = 0.880.
   $$

2. **Second Scenario: Probability of At Least 6 Out of 9 Elements Being Correct**

   Each element is correct with a probability of 0.7. Let $X \sim \text{Binomial}(9, 0.7)$, representing the number of correct elements. The probability of at least 6 elements being correct is calculated as:

   $$
   P_2 = P(X \geq 6) = \sum_{k=6}^{9} \binom{9}{k} (0.7)^k (0.3)^{9 - k} = 0.729.
   $$


So the probability that **at least 2 out of 3 group labels are correct** ($P_1 \approx 0.880$) is clearly greater than the probability that **at least 6 out of 9 elements are correct** ($P_2 \approx 0.729$).

**Withdrawal Confirmation:**

I have read and agree with the venue's withdrawal policy on behalf of myself and my co-authors.